# LHX2 Enhances the Malignant Phenotype of Esophageal Squamous Cell Carcinoma by Upregulating the Expression of SERPINE2

**DOI:** 10.3390/genes13081457

**Published:** 2022-08-16

**Authors:** Xukun Li, Xueling Wu, Hongyan Chen, Zhihua Liu, Huan He, Luhua Wang

**Affiliations:** 1Department of Radiation Oncology, National Cancer Center/National Clinical Research Center for Cancer/Cancer Hospital & Shenzhen Hospital, Chinese Academy of Medical Sciences and Peking Union Medical College, Shenzhen 518116, China; 2The State Key Laboratory of Molecular Oncology, National Cancer Center/National Clinical Research Center for Cancer/Cancer Hospital, Chinese Academy of Medical Sciences and Peking Union Medical College, Beijing 100021, China

**Keywords:** ESCC, LHX2, oncogene, tumor progression, SERPINE2

## Abstract

LHX2 dysregulations have been found to present in cancers, but the function of LHX2 in esophageal squamous cell carcinoma (ESCC) remains unknown. Here, we report that *LHX2* was upregulated in ESCC tissues in comparison to the *LHX2* levels in adjacent normal tissues. Loss- and gain-of-function experiments demonstrated that the knockdown of LHX2 markedly inhibited ESCC cells’ proliferation, migration, invasion, tumor growth and metastasis, whereas the overexpression of LHX2 had the opposite effects. A mechanistic investigation revealed that LHX2 bound to the promoter of *SERPINE2* gene and transcriptionally regulated the expression of SERPINE2. Collectively, LHX2 facilitates ESCC tumor progression, and it could be a potential therapeutic target for ESCC.

## 1. Introduction

Esophageal cancer is one of the most malignant cancers. In 2020, there were an estimated 604,000 new cases and 544,000 deaths worldwide [1]. This type of cancer can be divided into two histological categories: esophageal squamous cell carcinoma (ESCC) and esophageal adenocarcinoma (EAC). The geographical incidence of esophageal cancer varies broadly, in China, appropriately 90% of esophageal cancers patients have ESCC. As ESCC does not present obvious symptoms, only 20% of ESCC patients are diagnosed in the early stage, advanced or metastatic ESCC patients accounts for more than 40% of all esophageal cancer patients and the 5-year survival rate of ESCC patients is under 30% [2]. Although omics studies have elucidated the pathogenesis of ESCC, no dominant driver genes for ESCC tumorigenesis have been identified, nor have any specific targeted drugs been approved for use in ESCC therapy. Therefore, the molecular mechanisms underlying ESCC tumorigenesis and progression should be fully elaborated so that novel and effective strategies can be developed for the diagnosis and treatment of ESCC.

LIM-homeodomain gene 2 (*LHX2*), known to be an important transcription factor, belongs to the subfamily of homeobox genes which are specifically characterized by N terminal two tandem cysteine-rich zinc finger LIM domains for protein binding (cofactors) and a C terminal one homeodomain for DNA binding [3,4]. Numerous studies have reported that LHX2 is fundamental to embryonic development; it is involved in skin, nervous system, brain, eye and gonad development and hematopoiesis [5]. During the process of embryonic development, LHX2 exhibits multifarious phase- and cell type-specific functions by controlling signal transduction, cell fate determination and cell differentiation [3,6]. In addition, LHX2 is closely related to tumorigenesis and tumor progression. For example, it was found that LHX2 promotes breast cancer oncogenesis by increasing the PDGF-B expression in breast cancer cells [7]. In osteosarcoma, it was shown that LHX2 enhances cell proliferation and metastasis by activating the mTOR pathway and suppressing autophagy [8]. However, its possible functions in ESCC remain poorly understood.

In our study, firstly, we examined the expression level of *LHX2* in ESCC tissues and adjacent normal tissues. In vitro and in vivo functional experiments demonstrated that the knockdown of LHX2 significantly inhibited ESCC cells’ proliferation, tumor growth, migration, invasion, and metastasis, while the overexpression of LHX2 exerted the opposite effects. Moreover, mechanistic exploration showed that LHX2 augmented the malignant phenotypes of ESCC cells by upregulating the expression of SERPINE2. Overall, our findings suggest that LHX2 plays an oncogenic role in ESCC progression.

## 2. Materials and Methods

### 2.1. Cell Lines and Culture

The KYSE series of human ESCC cells was generously gifted by Dr. Yutaka Shimada (Kyoto University, Japan). Mouse ESCC mEC25 cells were kindly provided by Dr. Fu Li (Shenzhen University School of Medicine, China) [9]. HEK293T cells were bought from the American Type Culture Collection (ATCC). KYSE series cells were maintained in RPMI1640 medium supplemented with 10% fetal bovine serum (FBS). mEC25, HNM007 and HEK293T cells were grown in DMEM medium with 10% FBS. Cells were cultivated at 37 °C in a 5% CO_2_ incubator and were periodically detected with mycoplasma contamination and identified via short tandem repeat (STR) analysis.

### 2.2. Plasmids, Virus Package and Infection

shRNA oligonucleotides targeting *LHX2* gene (Table 1) were constructed into a pSIH-H1 vector, coding regions of mouse *Lhx2* and human *SERPINE2* were cloned into a pLVX-IRES-NEO vector, and coding regions of human *LHX2* were inserted into a pcDNA3 vector. HEK293T cells were seeded into 10 cm culture plates at 90% confluence and co-transfected with a lentivirus packaging system (pMD2.G and psPAX2) and pSIH-H1-shLHX2, pLVX-IRES-NEO-*Lhx2* or pLVX-IRES-NEO-*SERPINE2* plasmids using Hieff Trans™ Liposomal transfection reagent (40802ES02, Yeasen Biotechnology, Shanghai, China) and Opti-MEM medium (31985070, Thermo Fisher Scientific, Waltham, MA, USA). After 48 h, virus supernatant was collected, centrifuged for 20 min at 3000 rpm in a refrigerated centrifuge, and filtered with a 0.45 μm syringe filter. The 2 × 10^5^ ESCC cells were planted into a 6-well plate and infected with lentivirus for 48 h with the addition of polybrene (10 μg/mL). Positive cells were screened with 550 μg/mL G418 or 1 μg/mL puromycin for one week.

### 2.3. Western Blotting

Cells were lysed using the appropriate volume of RIPA lysis strong buffer added with protease inhibitor on ice for 20 min, followed by ultrasonic treatment to obtain the transparent protein supernatant. A BCA protein assay kit was employed to measure the protein concentration (23225, Thermo Fisher Scientific, Waltham, MA, USA). Equivalent proteins were sequentially loaded into 10% SDS-PAGE gel, separated with electrophoresis apparatus and transferred onto a PVDF membrane. The membranes were blocked with 5% skim milk at room temperature for 2 h, incubated with primary antibodies (LHX2, abcam, ab184337; SERPINE2, Santa Cruz, sc-365650; β-actin, A5316, Sigma, Louis, MO, USA) at 4 °C overnight, washed four times with TBST for 10 min each time, incubated with secondary antibodies at room temperature for 2 h and washed with TBST again. Finally, all the membranes were detected using an enhanced chemiluminescent substrate kit (34095, Thermo Fisher Scientific) and ImageQuant LAS 4000 system (GE Healthcare, Marlborough, MA, USA).

### 2.4. RNA Isolation and Quantitative Real-Time PCR

RNA was extracted from ESCC cells, ESCC tissues and matched adjacent normal tissues with TRIzol. cDNA was obtained using the reverse transcription kit (KR116, TIANGEN, Beijing, China). SYBR Green Master Mix was utilized to execute quantitative real-time PCR (A25742, Applied Biosystems, Carlsbad, CA, USA). Comparative Ct method were used to analyze the data, and β-actin served as a control. Then, 34 paired ESCC tissues and matched adjacent normal tissues were collected from ESCC patients (who had not undergone treatment) in the Cancer Hospital, Chinese Academy of Medical Sciences. Informed consent was obtained from all the patients, and this study was approved by the Institutional Review Board of the Cancer Hospital, Chinese Academy of Medical Sciences. The primers are listed in Table 1.

### 2.5. CCK-8 Assays and Colony-Formation Ability

For the CCK-8 assays, 2 × 10^3^ (KYSE30, KYSE510, mEC25 and HNM007) cells were plated into 96-well culture plates with four repeat wells in each group. Once cells had adhered, the culture medium was removed. CCK-8 reagent (C0005, TargetMoI, Shanghai, China) and fresh serum-free medium were added into the plates at a ratio of 1:10, and the plates were incubated at 37 °C for 1 h. The absorbance was determined using a microplate reader at 450 nm for 0, 2 and 4 d; then, a cell proliferation curve was drawn depending on the absorbance value. For the colony formation assays, 2 × 10^3^ ESCC cells were inoculated into 6-well plates at a single-cell density. Fresh medium was replaced every 2 days, and the colonies were cultured continuously for 8–10 days. The cell colonies were fixed with methanol and stained using 0.5% crystal violet. After washing them with PBS, the tinctorial colony number was calculated.

### 2.6. Transwell and Wound-Healing Assays

For the Transwell assays, 5 × 10^4^ KYSE30, 1.5 × 10^5^ KYSE510, 5 × 10^4^ mEC25 and 3 × 10^4^ HNM007 cells resuspended with 100 μL serum-free medium were planted into the upper chambers of 24-well Transwell units (3422, Corning, Corning, NY, USA) pre-embedded with Matrigel for invasion assays and without Matrigel for migration assays, and complete medium (10% FBS) was added into the bottom chambers. Twenty-four hours later, the migrated and invaded cells were fixed with methanol, stained with 0.5% crystal violet and counted within four random fields. For the wound-healing assays, the cells were seeded into 6-well plates at a density of 100% confluence. The adherent cells were scratched with a 200 µL pipette tip to form a gap and washed with PBS to remove cell debris, and fresh medium was added to the plates. Wound closure was monitored, and the representative images were taken at 0, 24 and 48 h.

### 2.7. Animal Experiments

For the subcutaneous transplantation experiments, 6-week-old male and female BALB/c nude mice were subcutaneously injected with 1 × 10^6^ KYSE30 and 5 × 10^6^ KYSE510 cells, respectively. The 4 × 10^6^ mEC25 and 1 × 10^5^ HNM007 cells were subcutaneously transplanted into 6-week-old male C57BL/6J mice. After almost a week, the length and width of tumors were measured for the first time. Tumor growth was constantly monitored and measured twice a week, and tumors were harvested 3–4 weeks after the injection. All tumors were harvested and tumor images were taken at the last measurement. Tumor volumes were analyzed based on volume = (length × width^2^)/2. For the experimental metastasis analyses, 1 × 10^6^ KYSE30 cells were injected into the tail veins of 6-week-old male SCID beige mice. After 3 months, SCID beige mice were intraperitoneally injected with D-luciferin (122799, Perkin Elmer, 3.75 mg/per mouse). The fluorescence value was detected using an IVIS Lumina XRMS system (PerkinElmer, Waltham, MA, USA) and lung tissues of these mice were dissected. The 1 × 10^5^ HNM007 cells were injected into the tail veins of 6-week-old male C57BL/6J mice, lung tissues of these mice were dissected at 1 month. The lung tissues were fixed with 4% paraformaldehyde for one week, stained using picric acid and the number of metastatic nodules was counted. All animal studies were approved by the Institutional Animal Care and Use Committee of Cancer Hospital, Chinese Academy of Medical Sciences.

### 2.8. RNA-Seq and ChIP Assays

RNA-seq and ChIP assays were performed as previously described [10]. Gene Ontology was analyzed for the differentially expressed genes; the threshold was defined as FDR < 0.05. KYSE30 cells were transfected with pcDNA3 or pcDNA3-*LHX2*-Flag plasmids with Hieff Trans™ Liposomal transfection reagent and Opti-MEM medium for 60 h, followed by ChIP assays with Flag antibody (14793, CST, Danvers, MA, USA) and SimpleChIP^®^ Enzymatic Chromatin IP Kit (9003, CST) according to the standard protocol. The binding of LHX2 to the promoter of the *SERPINE2* gene was determined via qPCR. The primers that were used are listed in Table 1.

### 2.9. Statistical Analysis

The experimental data were analyzed and illustrated using GraphPad Prism 9. Two tailed independent Student’s *t* tests were performed to determine statistical significance. The Spearman correlation was utilized to assess the association between the expression of two genes. Kaplan–Meier plots were used to visualize survival curves, and the significance between two groups was compared via log-rank analysis. Data are exhibited as mean ± SD, and *p* < 0.05 was considered to be statistically significant.

## 3. Results

### 3.1. LHX2 Is Significantly Upregulated in ESCC

The Kaplan–Meier survival analysis of Pan-cancer showed that patients with high *LHX2* expression had a shorter overall survival and disease-free survival when compared with patients with low *LHX2* expression (Figure 1A). Consistent with function of LHX2 in cancers in the previous studies [8,11,12,13], we consider that LHX2 mainly performs a tumor-promoting function in cancers. To determine the role of LHX2 in ESCC, firstly, we analyzed the available TCGA (The Cancer Genome Atlas) and GEO (Gene Expression Omnibus) ESCC datasets, and we found that the mRNA levels of *LHX2* were remarkably upregulated in ESCC tissues compared with the levels in normal tissues or matched adjacent normal tissues (Figure 1B,C). Meanwhile, our unpublished RNA-seq data of 155 paired ESCC tissues and adjacent normal tissues also indicated that ESCC tissues displayed much higher *LHX2* mRNA expression than matched adjacent normal tissues (Figure 1D). In line with these findings, we collected 34 paired ESCC tissues and adjacent normal tissues and performed the qRT-PCR experiments, the results further confirmed that the *LHX2* mRNA expression was increased in ESCC tissues in contrast to the expression in adjacent normal tissues (Figure 1E). Taken together, these findings suggest that LHX2 is upregulated in ESCC tissues.

### 3.2. LHX2 Enhances the Proliferation and Tumor Growth of ESCC Cells

To verify whether LHX2 executes the oncogenic functions in ESCC progression, firstly, we built the stable LHX2-knockdown KYSE30/KYSE510 cells with high endogenous levels of LHX2 expression and LHX2-overexpressing mEC25/HNM007 cells with low basal levels of LHX2 expression. The efficiency of LHX2 knockdown and overexpression was examined via qRT-PCR and Western blotting (Appendix A). Next, we explored the effect of LHX2 on the proliferation of ESCC cells by CCK-8 and colony formation assays, and discovered that the knockdown of LHX2 noticeably attenuated the proliferation and colony formation of KYSE30 and KYSE510 cells (Figure 2A–D), while the overexpression of LHX2 significantly promoted the proliferation and colony formation of mEC25 and HNM007 cells (Figure 2E–H). These results support the concept that LHX2 increased ESCC cell proliferation in vitro. To validate these findings in vivo, we subcutaneously injected LHX2-knockdown KYSE30/KYSE510 cells, LHX2-overexpressing mEC25/HNM007 cells and their corresponding control cells into BALB/C null and C57BL/6J mice. As expected, LHX2 silencing dramatically suppressed the tumor growth of ESCC cells (Figure 2I,J, Appendix A), and LHX2 overexpression facilitated tumors with greater volumes and heavier weights with a higher proliferation rate (Figure 2K,L, Appendix A). Overall, these data strongly suggest that LHX2 accelerates ESCC cell proliferation and tumor growth.

### 3.3. LHX2 Promotes the Migration, Invasion and Metastasis of ESCC Cells

Metastasis is the leading cause of cancer-associated deaths in ESCC patients; therefore, the identification of key genes related to metastasis would aid the development of novel therapeutic methods [14]. Next, we investigated the function of LHX2 in ESCC cells’ movement using Transwell and wound-healing assays. The results showed that LHX2-knockdown KYSE30 and KYSE510 cells were much less capable of cell movement than their corresponding control cells (Figure 3A,B,E, Appendix A). In contrast, the overexpression of LHX2 markedly accelerated the migration, invasion and wound-healing of mEC25 and HNM007 cells (Figure 3C,D,F, Appendix A). These results show that LHX2 is potentially involved in ESCC metastasis. To test this assumption, we injected KYSE30 and HNM007 cells into SCID beige and C57BL/6J mice via the tail veins, and then, we observed and counted the number of micrometastases in the lung tissues of mice. Consistent with the in vitro results, LHX2 knockdown notably impaired the pulmonary metastasis in SCID beige mice, whereas LHX2 overexpression strikingly increased the number of metastatic nodules in the lung tissues of C57BL/6J mice (Figure 3G–I). Collectively, we identified that *LHX2* acts as an oncogene in ESCC, and enhances ESCC cells’ motility in vitro and metastasis in vivo.

### 3.4. LHX2 Transcriptionally Regulates the Expression of SERPINE2 in ESCC

To explain how LHX2 enhances the malignant phenotype of ESCC, we performed transcriptome profiling using LHX2-knockdown KYSE30/KYSE510 cells and their corresponding control cells, and we identified that 71 genes were upregulated and 103 genes were downregulated in both KYSE30 and KYSE510 cells after *LHX2* silencing (Figure 4A,B; Appendix A). In the Gene Ontology (GO) analysis, it was shown that the 174 differentially expressed genes were significantly enriched in the regulation of signaling, the regulation of cell differentiation, the response to wounding, cell proliferation, cell migration and cell motility (Figure 4C). In terms of the reinforced effect of LHX2 on the proliferation and motility of ESCC cells, genes located in the GO terms of cell development, cell proliferation and cell motility were chosen for the following research. In addition, we next performed the RT-qPCR analysis to validate the changes in gene expression in KYSE30 and KYSE510 cells after the reduction in *LHX2*. The results demonstrated that the inhibition of *LHX2* obviously reduced the expressions of genes in the above three GO terms in both KYSE30 and KYSE510 cells, such as *MAPK6*, *SERPINE2*, *FGFBP1* and *GBP1* (Figure 4D,E). Meanwhile, we found that *SERPINE2* was significantly increased in ESCC tissues compared with adjacent normal tissues by analyzing the GEO datasets (Figure 4F), and Pearson correlation analysis showed that *SERPINE2* expression was positively correlated with *LHX2* expression in ESCC tissues (Figure 4G). Moreover, Western blotting confirmed that the knockdown of LHX2 obviously diminished the expression of SERPINE2 in KYSE30 and KYSE510 cells (Figure 4H). Given the possibility of a transcriptional regulatory relationship between *LHX2* and *SERPINE2*, so we selected *SERPINE2* gene as the candidate downstream target of *LHX2* transcriptional regulation. To ascertain whether LHX2 directly modulates the expression of SERPINE2, we transfected pcDNA3-*LHX2*-Flag plasmids into KYSE30 cells and performed the ChIP-qPCR assays using Flag antibody. The results revealed that LHX2 directly bound to the promoter region of the *SERPINE2* gene and transcriptionally regulated the expression of SERPINE2 (Figure 4I, Appendix A). Taken together, these findings suggest that LHX2 may enhance tumor growth and metastasis by transcriptionally upregulating the SERPINE2 expression in ESCC.

### 3.5. LHX2 Augments the Malignant Phenotype of ESCC by Upregulating SERPINE2

To further assess whether SERPINE2 exerts an essential effect on LHX2-induced cell proliferation and movement, we restored the expression of SERPINE2 in KYSE30 and KYSE510 cells with LHX2 knockdown, and the efficiency of LHX2 and SERPINE2 expression was evaluated via Western blotting and RT-qPCR (Figure 5A, Appendix A). Functional recovery experiments showed that the exogenous overexpression of SERPINE2 appreciably regained cell proliferation weakened by LHX2 knockdown in KYSE30 and KYSE510 cells via the CCK-8 and colony-formation assays (Figure 5B,C). Similarly, Transwell and wound-healing assays also displayed that the attenuated migration, invasion and wound-healing induced by LHX2 inhibition can be rescued to an extent through increasing the expression of SERPINE2 (Figure 5D,E, Appendix A). These results indicate that LHX2 promotes the malignant phenotype of ESCC at least partially by increasing the expression of SERPINE2. Taking these results together, it can be determined that SERPINE2 is a key functional target of LHX2-promoting cell proliferation and motility.

## 4. Discussion

As ESCC progresses, the abnormally accelerated proliferation of esophageal epithelial cells in the early stage is morphologically followed by basal cell hyperplasia, dysplasia and carcinoma in situ [15]. The available evidence shows that ESCC tumorigenesis and progression can mainly be attributed to inordinate cell proliferation, differentiation and cell death, and squamous differentiation is the crucial pathogenic mechanism that enables ESCC heterogeneity and malignant progression. Multiple-omics high-throughput sequencing has been used to identify several driver genes in ESCC, for example, mutations, deletions and amplifications of *TP53*, *ZNF750*, *NOTCH1*, *NFE2L2*, *TP63* and *SOX2* genes have frequently been identified in ESCC specimens, constituting the most common genomic features of ESCC [16,17]. However, these genes have not yet been developed into effective targets for ESCC treatment; more basic research is still necessary to obtain driver genes for targeted therapy toward ESCC.

As a homeobox gene, *LHX2* is a vital developmental regulator for growth and differentiation [18,19,20,21]. Many studies have revealed that aberrant proliferation and differentiation caused by the gain and loss of homeobox genes motivates tumorigenesis and tumor progression. Time- and tissue-specific expression patterns of homeobox genes maintain a balance between proliferation and differentiation in normal tissues. Nevertheless, disruptions to these gene expression patterns facilitate the development of malignantly transformed phenotypes [22,23,24]. LHX2 dysregulations have been detected in several cancers; the DNA methylation of *LHX2* provided the potential value by serving as a novel biomarker in cervical cancer radiotherapy [13], but the role of LHX2 in ESCC remains ambiguous. In our study, we found that the level of *LHX2* was increased considerably in ESCC tissues compared with adjacent normal tissues, and it was shown that LHX2 overexpression promoted tumor growth and metastasis, whereas LHX2 knockdown lessened the growth and metastasis of ESCC. Similar to our results, it was previously shown that LHX2 silencing weakened the migration and invasion of non-small cell lung cancer (NSCLC) [25], and LHX2 overexpression promoted cell proliferation by activating β-catenin/TCF signaling in pancreatic ductal adenocarcinoma [12]. Inversely, as a tumor suppressor, LHX2 inhibits tumor development by blocking the MAPK/ERK and Wnt/beta-catenin signaling in adult and pediatric liver cancers [26]. Additionally, LHX2 has also been recognized as a novel potential biomarker of lung carcinoids with invasive characteristics [27]. Combined with our findings, these results remind that LHX2 could be useful as a possible target for ESCC therapies. Regrettably, due to lack of developed inhibitors of LHX2, we failed to perform the therapeutic experimental studies to validate whether LHX2 has the potential valuable in ESCC treatment.

The mechanistic study showed that LHX2 bound to the promoter region of *SERPINE2* and directly transcriptionally regulated SERPINE2. Meanwhile, the functional recovery experiment proved that SERPINE2 is the key functional target of LHX2 in the promotion of ESCC malignancies, and *SERPINE2* expression was positively associated with *LHX2* expression in ESCC tissues. SERPINE2, a serine or cysteine proteinase inhibitor, is widely implicated in tumorigenesis and tumor progression [28]. *SERPINE2*, which was activated by oncogenic *Ras*, *BRAF* and *MEK1*, led to the ERK pathway playing a pro-neoplastic role in intestinal epithelial cells [29]. The mouse model of breast tumor heterogeneity demonstrated that SERPINE2 provokes tumor cells to form vascular networks that mimic blood vessels to facilitate tumor cells penetration and metastasis, indicating that *SERPINE2* is a dominant driver of metastatic progression [30]. Simultaneously, high SERPINE2 expression was an independent poor prognostic factor in oral squamous cell carcinoma (OSCC), lung adenocarcinoma, urothelial carcinoma and endometrial cancer [31,32,33,34], and SERPINE2 was identified as a promising therapeutic target in melanoma metastasis and in the radio-resistance of lung cancer. SERPINE2 promotes radio resistance by mediating the interaction of MRE11 with ATM to induce ATM phosphorylation in homologous recombination (HR)-mediated double strand break (DSB) repair [28,35]. In addition, as a secretory protein, the increased serum concentration of SERPINE2 was considered to be an additional marker for the differentiation of malignancies [36]. A previous study reported that SERPINE2 boosts ESCC metastasis by activating BMP4 [37]. As a transcription factor, LHX2 may transcriptionally regulate the expression of a group of tumor-related genes, SERPINE2 did not completely rescue the phenotype induced by LHX2 knockdown in ESCC, we speculate that there may be other gene mediating the roles of LHX2 in the malignant phenotypes. However, how SERPINE2 mediates LHX2-promoting tumor growth and the metastasis of ESCC still remains unclear, and our study failed to clarify the clinical significance of LHX2 and SERPINE2 in ESCC. In the future, more mechanistic research and clinical studies will be needed to fully elucidate the effect of LHX2 and SERPINE2 on the pathogenesis of ESCC.

## 5. Conclusions

In summary, LHX2 is significantly upregulated in ESCC tissues and promotes ESCC tumor growth and metastasis by augmenting the expression of SERPINE2. Our study demonstrates that LHX2 acts as a tumor-promoting role in ESCC progression and provides a new potential target against ESCC.

## Figures and Tables

**Figure 1 genes-13-01457-f001:**
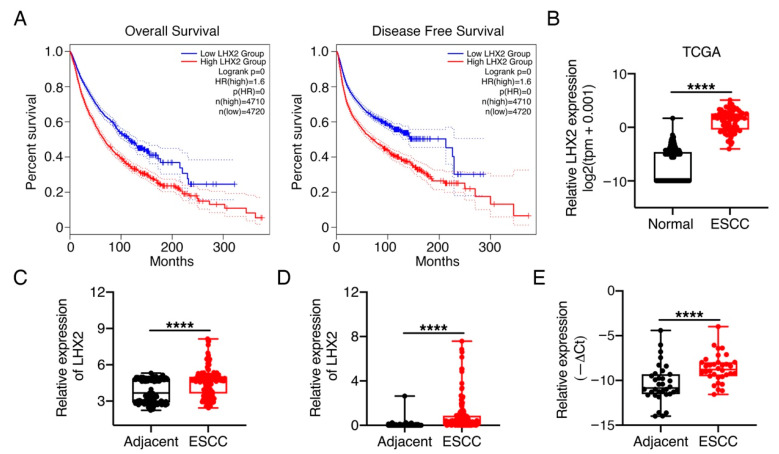
LHX2 is significantly upregulated in ESCC. (**A**) Kaplan–Meier analysis of overall survival and disease-free survival rates of Pan-cancer patients in TCGA dataset, patients were divided into two groups based on the median of *LHX2* expression (GEPIA 2). (**B**) The mRNA levels of *LHX2* were compared between the normal esophagus tissues (*n* = 273) and ESCC tissues (*n* = 75) by analyzing TCGA dataset. (**C**) The *LHX2* mRNA levels were evaluated in ESCC tissues and matched adjacent normal tissues by analyzing GEO datasets (GSE20347, GSE23400 and GSE44021, *n* = 183). (**D**) The *LHX2* mRNA levels were examined in 155 paired ESCC tissues and their matched adjacent tissues by RNA-seq. (**E**) The mRNA levels of *LHX2* were detected in 34 paired ESCC tissues and matched adjacent normal tissues via qRT-PCR. Data are presented as the mean ± SD, two-tailed *t*-tests, **** *p* < 0.0001.

**Figure 2 genes-13-01457-f002:**
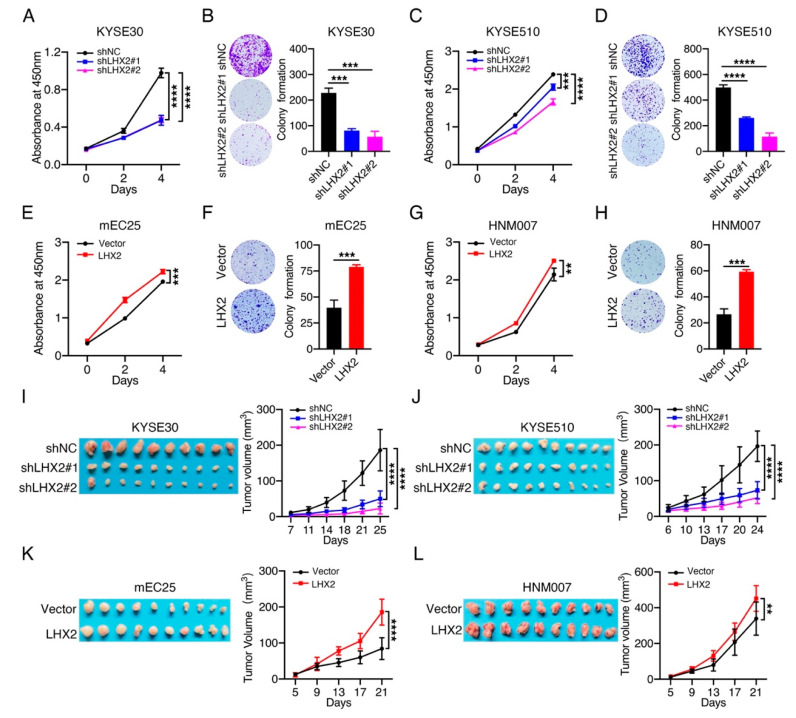
LHX2 promotes ESCC cell proliferation and tumor growth. (**A**,**C**,**E**,**G**) ESCC cell proliferation was determined using CCK-8 assays after the knockdown of LHX2 in KYSE30 and KYSE510 cells, and ectopic overexpression of LHX2 in mEC25 and HNM007 cells. (**B**,**D**,**F**,**H**) ESCC cell proliferation was analyzed using colony formation assays after the knockdown of LHX2 in KYSE30 and KYSE510 cells, and ectopic overexpression of LHX2 in mEC25 and HNM007 cells. Representative images of cell colonies (**left**) and the number of cell colonies (**right**) are presented. (**I**–**L**) LHX2-knockdown KYSE30/KYSE510 cells, LHX2-overexpressing mEC25/HNM007 cells and their corresponding control cells were subcutaneously planted into BALB/C null and C57BL/6J mice, respectively. Tumor growth was measured twice a week, tumors were harvested 3–4 weeks after the injection and tumor images were taken at the last measurement. Representative tumor images (**left**) and tumor growth curves (**right**) are shown (*n* = 10). Data are presented as the mean ± SD, two-tailed *t*-tests, ** *p* < 0.01, *** *p* < 0.001, **** *p* < 0.0001.

**Figure 3 genes-13-01457-f003:**
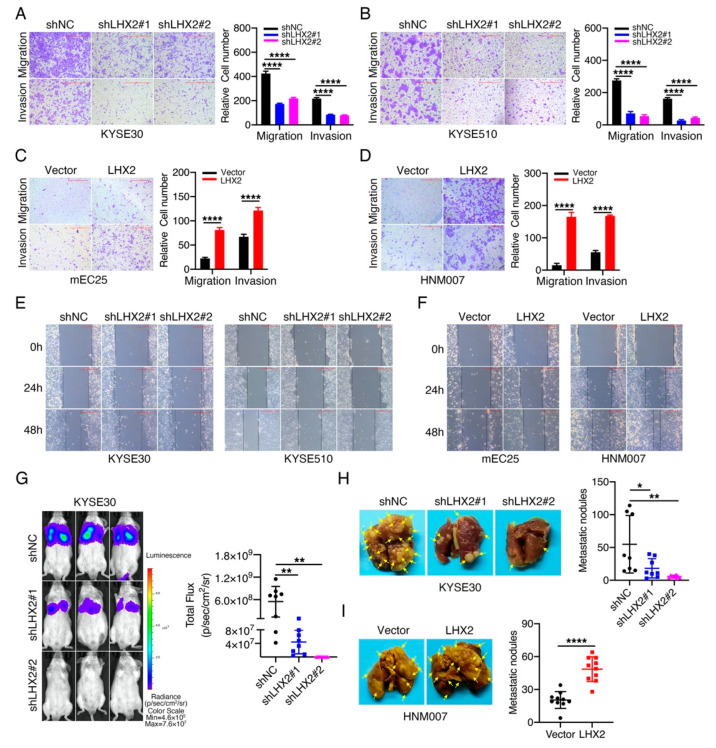
LHX2 facilitates ESCC cell migration, invasion and metastasis. (**A**–**D**) The ability of cell migration and invasion was determined using Transwell assays after the knockdown of LHX2 in KYSE30 and KYSE510 cells, and ectopic overexpression of LHX2 in mEC25 and HNM007 cells. Representative images (**left**) and the number of migrated and invaded cells (**right**) are shown. Scale bars = 500 μm. (**E**,**F**) The adherent cells were scratched with a pipette tip to form a gap. The wound closure was monitored at 0, 24 and 48 h. Representative images of wound-healing assays with LHX2-knockdown KYSE30/KYSE510 cells, LHX2-overexpressing mEC25/HNM007 cells and their corresponding control cells. Scale bars = 500 μm. (**G**) LHX2-knockdown of luciferase-labeled KYSE30 and their corresponding control cells were injected into SCID beige mice via tail vein. Mice was intraperitoneally injected with D-luciferin at 3 months and the fluorescence value was analyzed. Representative images of luciferase-labeled KYSE30 cells metastasized into lung tissues (**left**) and quantitative statistics of fluorescence value (**right**) are presented (*n* = 8). (**H**) LHX2-knockdown KYSE30 cells and their corresponding control cells were injected into SCID beige mice via the tail veins. Lung tissues of mice were dissected at 3 months and the number of metastatic nodules in the lung tissues were counted. Representative images (**Left**) and statistical results of metastatic nodules (**Right**) are shown (*n* = 8), and the yellow arrows point to the metastatic nodules. (**I**) LHX2-overexpressing HNM007 cells and their corresponding control cells were injected into C57BL/6J mice via the tail veins. Lung tissues of mice were dissected at 1 month and the number of metastatic nodules in the lung tissues were counted. Representative images (**left**) and statistical results of metastatic nodules (**right**) are shown (*n* = 10), and the yellow arrows point to the metastatic nodules. Data are presented as the mean ± SD, two-tailed *t*-tests, * *p* < 0.05, ** *p* < 0.01, **** *p* < 0.0001.

**Figure 4 genes-13-01457-f004:**
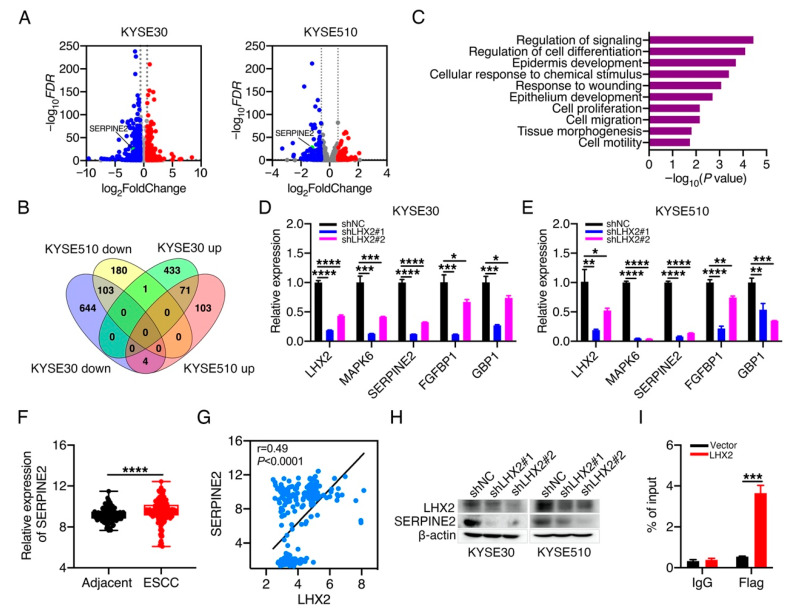
LHX2 transcriptionally regulates the expression of SERPINE2. (**A**,**B**) Volcano heatmap and Venn diagram showing the differentially expressed genes regulated by *LHX2* in KYSE30 and KYSE510 cells. (**C**) Gene Ontology analysis of 174 differentially expressed genes regulated by *LHX2* in both KYSE30 and KYSE510 cells; the length of bars shows significance. (**D**,**E**) qRT-PCR analyses were performed to detect the expression of *MAPK6*, *SERPINE2*, *FGFBP1* and *GBP1* in KYSE30 and KYSE510 cells after the knockdown of *LHX2*. (**F**,**G**) The mRNA levels of *SERPINE2* were analyzed in GEO dataset, and the Pearson correlation was carried out to determine the correlation between the *LHX2* expression and the *SERPINE2* expression in ESCC tissues (GSE20347, GSE23400 and GSE44021, *n* = 183). (**H**) Western blotting was used to detect the SERPINE2 expression in LHX2-knockdown KYSE30 and KYSE510 cells. (**I**) KYSE30 cells were transfected with pcDNA3-*LHX2*-Flag plasmids with transfection reagent for 60 h. Then ChIP assays were performed with Flag antibody and qPCR was applied to detect the binding of LHX2 at the promoter region of *SERPINE2* gene. Data are presented as the mean ± SD, two-tailed *t*-tests, * *p* < 0.05, ** *p* < 0.01, *** *p* < 0.001, **** *p* < 0.0001.

**Figure 5 genes-13-01457-f005:**
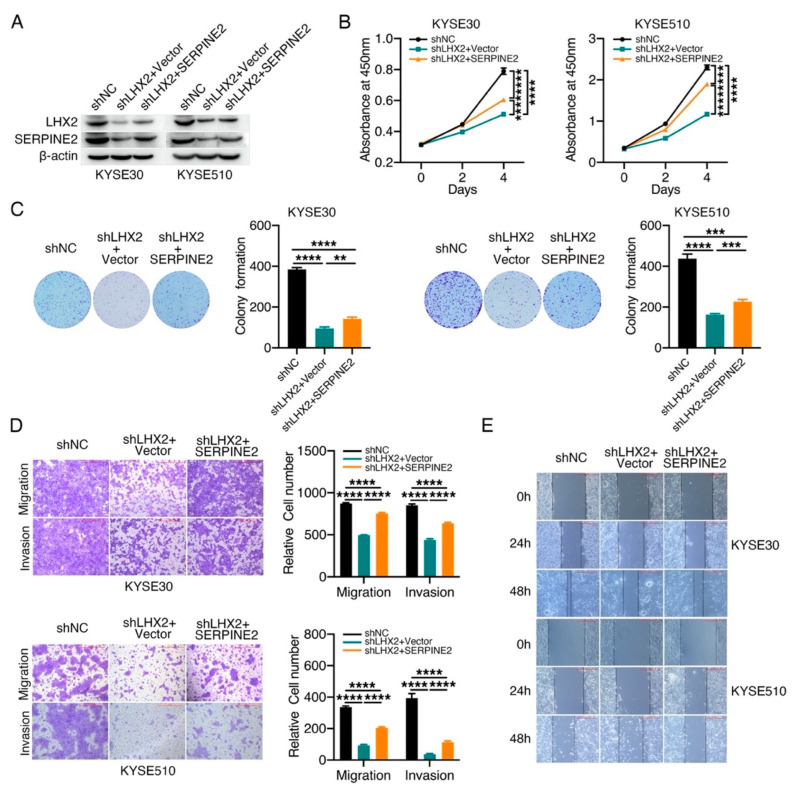
SERPINE2 is a functional target of LHX2. (**A**) The expression of SERPINE2 was detected via Western blotting in LHX2-knockdown KYSE30 and KYSE510 cells. The expression of SERPINE2 was restored by exogenous overexpression in LHX2-knockdown KYSE30 and KYSE510 cells. Then, (**B**) CCK-8, (**C**) colony formation, (**D**) Transwell and (**E**) wound-healing assays were performed to investigate the effect of exogenous overexpression of SERPINE2 on functional recovery in LHX2-knockdown KYSE30 and KYSE510 cells. Data are presented as the mean ± SD, two-tailed *t*-tests, ** *p* < 0.01, *** *p* < 0.001, **** *p* < 0.0001.

**Table 1 genes-13-01457-t001:** Primers used in this study.

Sequence	Used for
h*LHX2*-F, 5′-ACGCCAAGGACTTGAAGCAGCT-3′	qRT-PCR
h*LHX2*-R, 5′-TTTCCTGCCGTAAGAGGTTGCG-3′	qRT-PCR
h*β-actin*-F, 5′-AGGCACCAGGGCGTGAT-3′	qRT-PCR
h*β-actin*-R, 5′GCCCACATAGGAATCCTTCTGAC-3′	qRT-PCR
m*Lhx2*-F, 5′-GTCATCGACGAGATGGACCG-3′	qRT-PCR
m*Lhx2*-R, 5′-TGAAGCAGGTGAGTTCCGAC-3′	qRT-PCR
m*β-actin*-F, 5′-CATTGCTGACAGGATGCAGAAGG-3′	qRT-PCR
m*β-actin*-R, 5′-TGCTGGAAGGTGGACAGTGAGG-3′	qRT-PCR
h*SERPINE2*-F, 5′-AAGAAACGCACTTTCGTGGC-3′	qRT-PCR
h*SERPINE2*-R, 5′-GTGTGGGATGATGGCAGACA-3′	qRT-PCR
h*FGFBP1*-F, 5′-CAGGAGGAGGGCATCTCTCT-3′	qRT-PCR
h*FGFBP1*-R, 5′-TCCGGGCAACTTGTTTCCAA-3′	qRT-PCR
h*GBP1*-F, 5′-TAGCAGACTTCTGTTCCTACATCT-3′	qRT-PCR
h*GBP1*-R, 5′-CCACTGCTGATGGCATTGACGT-3′	qRT-PCR
h*MAPK6*-F, 5′-AGCTTGGGGAGAGAGGACAT-3′	qRT-PCR
h*MAPK6*-R, 5′-GTGGGATGCCTATGGACTCG-3′	qRT-PCR
*SERPINE2*-F, 5′-CATTTCTTAACAACTGCATGCCA-3′	ChIP-qPCR
*SERPINE2*-R, 5′-TCCCTGCATCCATTTCTGTCT-3′	ChIP-qPCR
5′-GCGCTAAGCTGCAACGAAA-3′	sh*LHX2*#1
5′-GCCAGAAGACCAAGCGCAT-3′	sh*LHX2*#2

## Data Availability

Transcriptome profiles of this study have been uploaded to GEO (GSE209942).

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
