# Peer review of "LHX2 Enhances the Malignant Phenotype of Esophageal Squamous Cell Carcinoma by Upregulating the Expression of SERPINE2"

_genes, 2022, doi:10.3390/genes13081457_

Round 1

Reviewer 1 Report

Li et. al. have performed a very extensive research to provide new insight into a tumor oncogenic function of LHX2 gene in esophageal squamous cell carcinoma (ESCC). Using publicly available resource, performing molecular assays such as RNA-seq, qRT-PCR, Chip-seq, proliferation & migration assay, and animal study, authors have done a lot of work to pin point the role of LHX2 in ESCC. Kudos to the authors for performing such an extensive study and I personally enjoyed reading this manuscript. 

I support publication of this manuscript after few minor changes. Please find my major and minor concerns below:

MAJOR CONCERN:

1. Authors don't show a very convincing link between LHX2 and SERPINE2. There is some evidence showing LHX2 might be enhancing ESCC malignancy via upregulating SERPINE2, however, I find this link the weakest among all the evidence presented in this study. In the rescue experiment by providing SERPINE2 back in LHX2 knockdown cells, the phenotype is very weak. 

If authors can show via some experiments (SERPINE k/d and phenotype study in some combination of +/- LHX2 gene) or discuss this limitation in the discussion section, it should suffice. 

MINOR CONCERN:

1. Line 33, the sentence should be neither/no instead of none.
2. In method 2.6 (Line119), can you discuss how did you distinguish between migrated and invaded cells? 
3.  Please write full form of TCGA and GEO in line 160.
4. Please discuss why did you use KYSE30/KYSE510 cells for knock down study and mEC25/HNM007 for over expression study.
5. In fig 2 I-L, did you organize the tumors according to size or is it days post plantation? If it's days post plantation, please include that information in the tumor images.
6. Line 213, I would use suggests not remind. 
7. Line 219, I would use enhances not strengthens.
8. Please include scale bar in Fig 3 A,b,c,d,e,f. 
9. If possible, quantification of gap closure in Fig 3 E-F, and 5E would be beneficial to the readers. 
10. In RNA-seq data, SERPINE2 is ranked 109 and 37 in 2 different cell lines. Is there a reason to dig into SERPINE2, please include that information as it was not clear. 

Reviewer 2 Report

Xukun Li et al. present an interesting study examining the role of LHX2 and downstream targets in esophageal squamous cell carcinoma tumor development.  The authors examined LHX2 expression through in vitro and in vivo models to explore its effects on cellular proliferation and metastatic potential.  The authors also examined gene expression profiles in cells with aberrant LHX2 expression to identify possible gene targets that may mediate its effects.  The role of SERPINE2 as a downstream effector of apparent LHX2 oncogenic properties was also explored.

Overall this is an interesting study.  The results presented show several different assays used to explore how LHX2 overexpression may contribute to tumor development or progression in ESCC. The raw data adds to the literature and provides important information to support a potential role for LHX2 in ESCC biology.  However, while this paper was interesting, there were some sections that were difficult to read due to grammar, requiring rereading to better understand the content.  Description of the result findings were often limited to brief interpretations requiring the reader to rely on figure data to see what the result was.  Figure legends also were insufficient in some instances to understand what the figures demonstrated.

Additional comments are outlined below:

Line 158 – while I agree that the Kaplan Meir curves (Figure 1) show improved overall survival and disease free survival for pan-cancer patients with low LHX2 versus high LHX2, this information alone is insufficient to conclusively demonstrate that “LHX2 mainly performs the tumor-promoting function in cancers”.  The authors have also not clarified criteria to differentiate, low versus high LHX2 in this data set. 

Line 167-168 – The data presented clearly shows enhanced LHX2 expression in ESCC tissues.  However, in tumor tissues enhanced gene expression can either modulate tumorgenic potential or be an indirect, secondary effect of other oncogenic processes driving tumorgenesis.  I agree with the statement that the gene expression data presented in Figure 1 suggests that LHX2 “may play a pro-oncogenic role in ESCC”, but wonder if the observed expression levels are directly or indirectly related to ESCC in these tissues. Some of the experiments described later in the manuscript help to build evidence supporting a role for LHX2 in advanced tumor biology, but the expression data on its own provides only weak evidence to support this role. 

Figure 2: The methods indicate that the CCK-8 was repeated in four wells for each test.  Are the graphs in Figure 2 a-h representative of all four wells from a single test, or from multiple replicates of the same test?  If the test was replicated, were similar significant differences observed between all test replicates?

Figure 2 I-L: Images of tumors are provided in figure 2 I-L, but it is not clear what day post implant that each of the represented tumor images are from, or whether these represent tumors from multiple days of experimentation post inoculation.

Lines 178-194 – If LHX2 is acting solely as a proto-oncogene in ESCC cells, then one would expect similar cellular proliferation in cells with the vector versus cells having LHX2 knocked out.  By contrast, the CCK-8 and colony formations showed decreased proliferation when LHX2 is knocked out.  This could be interpreted as a possible role for normal LHX2 in maintenance of cellular proliferation. Moreover, reduced proliferation of LHX2 knock out cells suggests LHX2 may be useful as a possible target for ESCC therapies. A possible role for LHX2 in maintaining cellular proliferation or targeted therapies is not discussed by the authors, but may be considered for further elaboration. 

Figure 3 H and I – It is not clear from the legend or text what the images are of in Figure H and I.  Are these the mice lungs, metastatic nodules or sites of tail injection? The images are too small to see what the arrows are pointing to.  There is also no descriptor provided about what the arrows indicate.

Lines 281-283 – The authors indicate that the results “strongly indicate that LHX2 promotes the malignant phenotype of ESCC by increasing the expression of SERPINE2” and that “SERPINE2 is a key functional target of LHX2-promoting cell proliferation and motility.” I agree that the results presented in Figure 5 support the hypothesis that SERPINE2 likely mediate growth effects of LHX2.  However, I also note from the data in Figure 5 that expression of SERPINE2 is insufficient to restore normal cell proliferation in cells with aberrant LHX2 expression. Elaboration on the failure to fully rescue the phenotype is suggested.

Minor comments:

Line 29-30 – Is this intended to mean that 90% of esophageal cancers in China are ESCC, or that 90% of ESCC worldwide occur in China?

Discussion – Lines 291 to 312 – Information from the first paragraph and a half appears more like introductory material that discussion of the research findings.

Thank you for the opportunity to review this interesting manuscript.
